# RCR: Relation-Centric Reasoning with Large Language Models for Knowledge-based Question Answering

## Abstract

While Large Language Models (LLMs) are powerful reasoners, their effectiveness on complex, knowledge-intensive tasks is fundamentally limited by their reliance on static, parametric knowledge, often leading to factual hallucinations. Augmenting them with external Knowledge Graphs (KGs) provides crucial factual grounding to address this limitation. Existing KG-based retrieval-augmented methods are predominantly based on triple-level retrieval, yet the vast and noisy entity space leads to suboptimal accuracy and scalability. This paper introduces RCR (Relation-Centric Reasoning), a new paradigm that pivots away from the vast entity space to the more stable and semantically richer space of relations. RCR first retrieves a compact set of candidate relations, then employs an LLM to compose them into abstract reasoning paths, and finally materializes these paths into a concrete evidence subgraph using the proposed similarity-based substitution mechanism to ensure robustness. On the challenging WebQSP and CWQ multi-hop question answering benchmarks, RCR achieves state-of-the-art performance. By prioritizing relations as the backbone of reasoning, RCR delivers a more accurate, interpretable, and scalable solution for KG-augmented reasoning.

## 1 Introduction

Large Language Models (LLMs) have demonstrated remarkable capabilities in natural language understanding, logical reasoning, and open-domain question answering (QA) (Zhao et al., 2023; Plaat et al., 2024; Bang et al., 2023; Brown et al., 2020). Despite these advances, their performance is often undermined by critical limitations, including reliance on stale internal knowledge (Petroni et al., 2019) and a tendency to factual hallucinations (Huang et al., 2025; Mündler et al., 2023). To solve these issues, a prominent way is to augment LLMs with external knowledge sources. Knowledge graphs (KGs) — structured repositories of entities and their relationships — have emerged as a particularly effective resource for enhancing the factual grounding and reasoning reliability of LLMs (Pan et al., 2024; Peng et al., 2024).

Within this domain, Knowledge Graph-based Retrieval-Augmented Generation (KG-RAG) has become a leading paradigm (Li et al., 2023a; Jiang et al., 2023b; Zhu et al., 2025; Wu et al., 2025). These methods retrieve structured triples from a KG to provide factual context for the LLM's reasoning process. However, the vast majority of existing KG-RAG frameworks are built upon a *triple-level scoring* approach (Li et al., 2024b; Yao et al., 2025), which suffers from two fundamental weaknesses. First, retrieving information based on entity mentions is highly susceptible to noise and ambiguity arising from entity linking errors (Ren et al., 2016; Chen et al., 2020; Hachey et al., 2013). Second, expanding subgraphs from seed entities often incorporates a large volume of redundant or irrelevant information (He et al., 2021), which can degrade reasoning quality and reduce efficiency. The sheer scale and complexity of entities in large KGs introduce significant challenges. For example, Freebase (Bollacker et al., 2008) comprises roughly tens of millions of entities but only tens of thousands of relations. Retrieving and reasoning over this vast and often ambiguous entity space is highly susceptible to noise and frequently incorporates redundant or irrelevant information.

In this paper, we argue that the relations in KGs, which are far less numerous and more semantically stable (Paulheim, 2016), offer a more robust and efficient foundation for reasoning. Motivated by

this, this paper introduces RCR, a novel framework that fundamentally shifts the paradigm from triple-level scoring to relation-centric path reasoning. Instead of navigating noisy and expansive entity neighborhoods, RCR identifies semantically meaningful relation paths, which provide an interpretable and compact scaffold for constructing task-relevant subgraphs.

The RCR framework operates through a multi-stage pipeline: it first retrieves candidate relations relevant to the question, then prompts an LLM to generate logical relation paths. These paths serve as an abstract blueprint to extract a compact subgraph, which is then used to generate a factually grounded final answer. Prior work has shown that many reasoning paths generated by LLMs are invalid or unsupported by the KG, leading to factual errors (Luo et al., 2024; Li et al., 2024a; Sui et al., 2024). RCR directly confronts this challenge. By employing a flexible similarity-based substitution mechanism and anchoring the reasoning process in valid relation paths, the framework ensures a strong alignment between the LLM's reasoning steps and the KG's factual structure.

Extensive evaluations on two standard KG-based QA benchmarks demonstrate the effectiveness of RCR. The framework achieves state-of-the-art performance while offering superior interpretability and robustness. These findings highlight the advantages of adopting a relation-centric perspective to address the challenges in KG-RAG systems.

In summary, the main contributions of this work are as follows:

- We propose RCR, a novel relation-centric framework that shifts the focus from noisy entity-based retrieval to robust relation path-guided reasoning, thereby enhancing the interpretability and robustness of KG-augmented LLMs reasoning.

- The design of a multi-stage architecture that seamlessly integrates dense relation retrieval, LLM-based path generation, and path-guided subgraph extraction to create a cohesive pipeline for structured knowledge utilization.

- The introduction of a similarity-based relation substitution mechanism that enhances the framework's robustness by allowing flexible adaptation to mismatches between predicted paths and the underlying KG structure.

## 2 RELATED WORK

### 2.1 LLM REASONING

Recent advancements in Large Language Models (LLMs) have demonstrated their powerful reasoning capabilities, which can be largely attributed to sophisticated prompting techniques and reasoning frameworks (Wei et al., 2022; Zhang et al., 2022; Wang et al., 2022; Yao et al., 2023; Wang et al., 2023a; Besta et al., 2024). For example, the Chain-of-Thought (CoT) (Wei et al., 2022), which explicitly encourages LLMs to break down complex reasoning tasks into intermediate reasoning steps before arriving at a final answer, significantly improves their performance on various reasoning benchmarks. Kojima et al. (2022) further shows that simple zero-shot prompts like "Let's think step by step," can elicit multi-step reasoning capabilities without requiring extensive fine-tuning or additional training data. Building upon the sequential reasoning paradigm of CoT, subsequent work has explored more complex reasoning structures. For instance, Tree-of-Thought (ToT) prompting (Yao et al., 2023) enables the model to explore multiple reasoning paths by organizing intermediate steps into a tree structure, thus facilitating more flexible and accurate multi-step reasoning. GoT (Besta et al., 2024) models the reasoning process as an arbitrary graph, allowing for the synthesis of arbitrary reasoning paths.

Despite these advancements, the reasoning capabilities of LLMs remain constrained by their internal, parametric knowledge. This knowledge may be outdated or incomplete, leading to factual inaccuracies and thus underscoring the necessity of augmenting LLMs with external knowledge sources.

### 2.2 KG-ENHANCED LLM

Due to the limitations mentioned above, recent research has explored integrating LLMs with external Knowledge Graphs (KGs). KGs provide structured, interpretable, and editable knowledge that

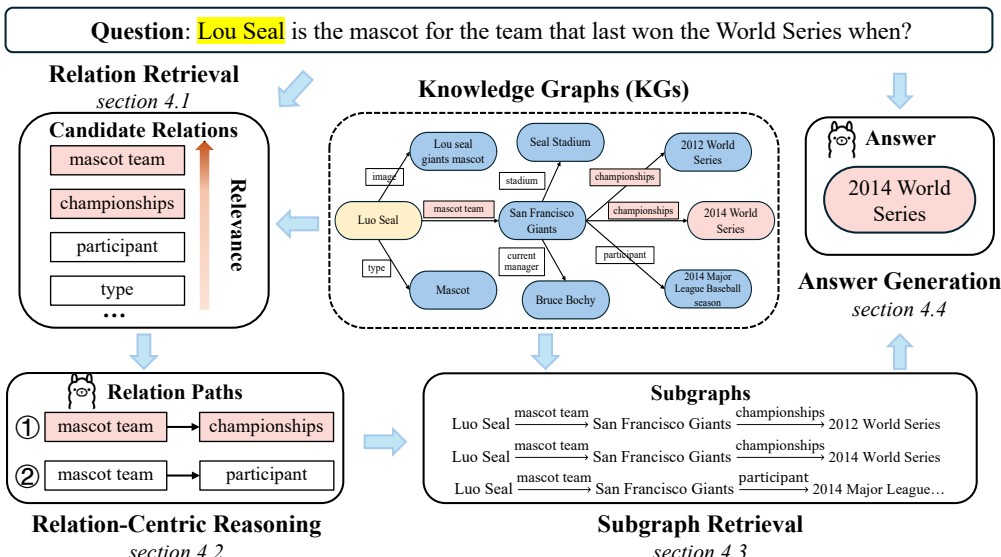

Figure 1: An overview of the proposed RCR framework. Given a natural language question, RCR first retrieves a set of candidate relation paths from the KG based on semantic similarity. Then, an LLM reasons over these paths to construct a coherent multi-hop reasoning path, which is used to retrieve relevant subgraphs. Finally, the LLM generates the answer based on the question and the retrieved subgraphs.

can ground the LLM's reasoning process, thereby improving factual accuracy and providing more interpretable outputs (Luo et al., 2024; Xu et al., 2024; Liu et al., 2024; Jiang et al., 2023b). Existing approaches can be broadly categorized into two paradigms: agent-based methods and retriever-based methods (Peng et al., 2024; Pan et al., 2024).

Agent-based approaches treat the LLM as an agent that iteratively interacts with the knowledge graph during reasoning to find the answer, enabling dynamic, step-by-step exploration of the KG. For instance, Think-on-Graph (ToG) (Sun et al., 2023; Ma et al., 2024) treats KG-based QA as an LLM-driven beam search over the graph. Beginning with topic entities, the LLM generates potential relation-entity paths to expand, maintaining a beam of the most promising reasoning chains. Similarly, Plan-on-Graph (PoG) (Chen et al., 2024) enhances this process by introducing a self-correcting planning module. It decomposes the primary question into sub-objectives, performs guided exploration, and utilizes a memory module with reflection to prune ineffective paths and backtrack when necessary. While these methods effectively leverage the structured nature of KGs, their iterative nature often leads to high latency and significant computational overhead, posing scalability challenges for large KGs (Mavromatis & Karypis, 2024; Li et al., 2024b).

Retriever-based approaches follow a Retrieval-Augmented Generation (RAG) paradigm (Lewis et al., 2020; Borgeaud et al., 2022; Vu et al., 2023; Fan et al., 2024), where relevant subgraphs are retrieved from KGs and then fed into the LLM for reasoning and answer generation. GNN-RAG (Mavromatis & Karypis, 2024) integrates the reasoning capabilities of graph neural networks (GNNs) with the understanding of LLMs. A GNN is used to reason over the subgraphs to score entities as answer candidates. Then, the shortest path between question entities and top-scoring entities is used to prompt the LLM as structured reasoning traces. SubgraphRAG (Li et al., 2024b) uses a lightweight MLP and triple-scoring mechanism to retrieve KG triples, with directional structural distances to better capture graph topology. This enables retrieving subgraphs tailored for multi-hop reasoning while keeping computation lightweight. However, these approaches often struggle to ensure that the retrieved triples can be integrated into a valid and coherent reasoning path (Long et al., 2025; Yao et al., 2025). Additionally, the abundance and noisiness of entities in KGs pose significant challenges. For example, these methods typically represent opaque entity identifiers (e.g., `m.48cu1`) as randomly initialized embeddings (Yao et al., 2025), which are not only difficult to interpret but may also introduce noise during the retrieval process.

In contrast, relations in KGs are generally more semantically meaningful and better aligned with natural language. They are far less numerous and more semantically stable than entities, providing a

more robust and efficient foundation for reasoning. This motivates our approach, RCR, which shifts the focus to relation-centric retrieval and path construction. By leveraging the semantic clarity of relations, RCR generates interpretable and coherent multi-hop reasoning paths. This method needs only two LLM calls during inference, making it efficient and scalable.

## 3 PRELIMINARIES

**Knowledge Graph (KG).** A Knowledge Graph is a structured representation of factual knowledge, formally defined as a set of triples $\mathcal{G} = (\mathcal{E}, \mathcal{R})$, which consists of a set of triples $\{(h, r, t) \mid h, t \in \mathcal{E}, r \in \mathcal{R}\}$. Here, $\mathcal{E}$ is a set of entities (nodes) and $\mathcal{R}$ is a set of relations (edges). Each triple $(h, r, t)$ represents a factual assertion, indicating that a head entity $h$ is connected to a tail entity $t$ via the relation $r$.

**Relation Reasoning** refers to the process of performing multi-hop inference over a knowledge graph by traversing a sequence of relations that semantically connect a topic entity to a potential answer. Formally, a relation path is represented as $\mathcal{P} = (r_1, r_2, \ldots, r_l)$, where each $r_i \in \mathcal{R}$ denotes a relation in the knowledge graph, and $l$ is the length of the path.

**Reasoning Path.** A Reasoning Path is a concrete instantiation of a Relation Path on the KG, forming a walkable trail from a topic entity to an answer entity. Given a starting entity $e_0$, a reasoning path is a sequence of entity-relation traversals $\mathcal{Z} = (e_0 \xrightarrow{r_1} e_1 \xrightarrow{r_2} \ldots \xrightarrow{r_l} e_l)$, where each step $(e_{i-1}, r_i, e_i)$ is a valid triple in $\mathcal{G}$.

## 4 PROPOSED METHOD

In this section, we propose RCR, a multi-stage framework for knowledge graph question answering. Unlike conventional KGQA pipelines that rely on triple-level scoring (Yao et al., 2025; Li et al., 2024b), RCR explicitly models the semantic structure of relations as the primary reasoning unit. This relation-centric paradigm enables more interpretable and efficient inference by first constructing abstract reasoning plans before grounding them in the KG. RCR decomposes the complex KGQA task into four sequential stages: relation retrieval, relation-centric reasoning, path-guided subgraph retrieval, and answer generation. The overall framework is illustrated in Figure 1.

### 4.1 RELATION RETRIEVAL

In the first stage, a lightweight relation retriever identifies a compact set of candidate relations semantically related to the question. Given a question $q$, the target of relation retriever $Q_\theta(q, \mathcal{R})$ is to identify a small and high-quality set of relevant relations $\mathcal{R}_q \subset \mathcal{R}$ from the KG. This set serves as the vocabulary for the subsequent path reasoning stage, drastically constraining the search space.

#### 4.1.1 RETRIEVER ARCHITECTURE

The relation retriever is designed as a dual-encoder model that maps both questions and relations into a shared semantic space. Following Li et al. (2024b), we first leverage off-the-shelf pre-trained text encoders to obtain initial fixed-length embeddings: the question $q$ is encoded as $\mathbf{q} \in \mathbb{R}^d$, and each relation $r \in \mathcal{R}$ is encoded as $\mathbf{r} \in \mathbb{R}^d$. To adapt these embeddings for relation retrieval, we introduce lightweight yet effective projection modules, $f_q(\cdot)$ and $f_r(\cdot)$, which are implemented as two-layer FFNs, to map the embeddings into a specialized retrieval space:

$$\tilde{\mathbf{q}} = f_q(\mathbf{q}), \quad \tilde{\mathbf{r}} = f_r(\mathbf{r}).$$

The similarity between a question and a candidate relation is then computed via cosine similarity:

$$\mathbf{sim}(q, r) = \cos(\tilde{\mathbf{q}}, \tilde{\mathbf{r}}). \tag{1}$$

During inference, all relation embeddings $\tilde{\mathbf{r}}$ are pre-computed and indexed, allowing for efficient top-$K_r$ retrieval based on maximum similarity.

### 4.1.2 TRAINING OBJECTIVE

To effectively train the retriever, we propose a contrastive learning objective tailored for relation retrieval. For each question $q$ in a training batch, we construct a set of positive relations $\mathcal{R}_q^+$, derived from the ground-truth reasoning paths, and a set of negative relations $\mathcal{R}_q^-$, obtained by random sampling. We then introduce a margin-based ranking loss designed to score positive relations higher than negative ones:

$$\mathcal{L} = \sum_{r^+ \in \mathcal{R}_{q_i}^+} \sum_{r^- \in \mathcal{R}_{q_i}^-} \max(0, \gamma - \mathbf{sim}(q_i, r^+) + \mathbf{sim}(q_i, r^-)) \tag{2}$$

where $\gamma$ is a margin hyperparameter and $\mathbf{sim}(\cdot, \cdot)$ is the cosine similarity. The loss is averaged over all triplets in the batch. This proposed objective effectively encourages the model to pull semantically relevant relations closer to the question in the shared embedding space while pushing irrelevant ones away, thereby improving retrieval accuracy.

### 4.2 RELATION-CENTRIC REASONING

With the candidate relation set $\mathcal{R}_q$ retrieved, the next challenge is to compose these relations into logically coherent multi-hop reasoning paths. Directly enumerating all possible compositions of relations in $\mathcal{R}_q$ leads to a combinatorial explosion in the search space. To address this, we harness the powerful reasoning capabilities of LLMs to synthesize interpretable reasoning paths. Compared to Luo et al. (2023) that directly predicts the reasoning paths given the question and topic entity, our framework first retrieves a small set of relevant relations to substantially constrain the search space, allowing for more accurate path generation.

#### 4.2.1 PATH GENERATION VIA SEQUENCE MODELING

We formulate this reasoning process as a relation-level sequence generation task. Formally, given a question $q$, its topic entity $e_0$, and the retrieved relation subset $\mathcal{R}_q \subseteq \mathcal{R}$, the LLM is prompted to generate plausible multi-hop relation paths $\mathcal{P} = (r_1, r_2, \ldots, r_l)$. This approach leverages the LLM's understanding of semantic compositionality to infer how relations should connect to form a valid reasoning chain from the topic entity towards a potential answer. The conditional probability of a path $\mathcal{P}$ is modeled autoregressively:

$$P(\mathcal{P} \mid q, e_0, \mathcal{R}_q) = \prod_{t=1}^{T} P(r_t \mid r_{<t}, q, e_0, \mathcal{R}_q) \tag{3}$$

#### 4.2.2 INSTRUCTION-TUNED GENERATION

To guide the LLM in generating structured paths, we fine-tune it on this task using an instruction-based format. This instruction tuning enhances the model's understanding of relation semantics and improves alignment with the KG structure. The model is trained on examples consisting of $(q, e_0, \mathcal{R}_q)$ as input and a ground-truth relation path $\mathcal{P}^*$ as the target output. We design a simple prompt template to instruct the LLM to generate relation paths:

> Please generate a valid relation path that connects the question entity to potential answer entities using only the provided candidate relations.
> Question: `<Question>`
> Question Entity: `<Entity>`
> Candidate Relations: (`<Relation 1>`, `<Relation 2>`, ..., `<Relation K>`)...

where `<Relation 1>`, `<Relation 2>`, ..., `<Relation K>` are the relations in $\mathcal{R}_q$. The candidate relations provided in the prompt consist of two parts: the relations in the ground-truth path and additional randomly sampled relations from the KG. The complete prompt template is provided in Appendix A.3. The training objective minimizes the negative log-likelihood of the target relation sequence:

$$\mathcal{L}_{\text{path}} = -\sum_{t=1}^{T} \log P(r_t^* \mid r_{<t}^*, q, \mathcal{R}_q) \tag{4}$$

Table 1: Performance comparison of RCR with baseline methods on WebQSP and CWQ datasets. Best results are highlighted in **bold**. Second-best results are underlined.

| Type | Methods | WebQSP | | CWQ | |
|---|---|---|---|---|---|
| | | Hits@1 (%) | F1 (%) | Hits@1 (%) | F1 (%) |
| Prompting (LLM Only) | Llama2-7B (Touvron et al., 2023) | 56.4 | 36.5 | 28.4 | 21.4 |
| | Llama3.1-8B (Meta, 2024) | 55.5 | 34.8 | 28.1 | 22.4 |
| | Qwen2-7B (Team, 2024) | 50.8 | 35.5 | 25.3 | 21.6 |
| | ChatGPT (OpenAI, 2022) | 59.3 | 43.5 | 34.7 | 30.2 |
| | ChatGPT+CoT (Wei et al., 2022) | 73.5 | 38.5 | 47.5 | 31.0 |
| | ChatGPT+Self-Consistency (Wang et al., 2022) | 83.5 | 63.4 | 56.0 | 48.1 |
| KG + LLM | UniKGQA (Jiang et al., 2022) | 77.2 | 72.2 | 51.2 | 49.1 |
| | ToG + ChatGPT (Sun et al., 2023) | 76.2 | - | 57.6 | - |
| | EtD + ChatGPT (Liu et al., 2024) | 82.5 | - | 62.0 | - |
| | G-Retriever (He et al., 2024) | 73.4 | 53.4 | - | - |
| | StructGPT (Jiang et al., 2023a) | 74.6 | - | - | - |
| | KD-CoT (Wang et al., 2023b) | 68.6 | 52.5 | 55.7 | - |
| | SubgraphRAG + Llama3.1-8B (Li et al., 2024b) | 86.6 | 70.5 | 56.9 | 47.2 |
| | RAPL + Llama3.1-8B (Yao et al., 2025) | 87.8 | **74.8** | 57.6 | 48.6 |
| | RoG (Luo et al., 2023) | 85.7 | 70.8 | 62.6 | 56.2 |
| | RCR | **88.9** | 69.3 | **66.7** | **57.3** |

At inference, we use beam search to generate a diverse set of candidate paths, capturing multiple potential lines of reasoning.

## 4.3 PATH-GUIDED SUBGRAPH RETRIEVAL

The abstract relation paths generated by the LLM represent reasoning hypotheses. This stage grounds these hypotheses in the KG by extracting a concrete subgraph. For each reasoning path $\mathcal{P}_i = [r_1, r_2, \ldots, r_T]$, we collect all triples $(e_{t-1}, r_t, e_t) \in \mathcal{G}$ that match the predicted relations. The retrieved subgraph is defined as:

$$\mathcal{G}_q = \bigcup_{\mathcal{P}_i} \left\{ (e_{t-1}, r_t, e_t) \in \mathcal{G} \mid e_{t-1} \in \mathcal{E}_{t-1}^{(i)}, \ r_t \in \mathcal{P}_i \right\} \qquad (5)$$

Here, $\mathcal{G}_q \subset \mathcal{G}$ denotes the task-specific subgraph retrieved by executing the paths $\mathcal{P}_i$ over the KG.

Although the LLM generates plausible reasoning paths, the predicted relations $r_i$ may not exist as an outgoing edge from the current entity $e_{i-1}$. To handle such cases and enhance robustness, we introduce a similarity-based relation substitution mechanism. If the exact relation $r_i$ is missing at a given step, we leverage our trained relation retriever (from Section 4.1) to find the top-$K_s$ relations $\{r_i^{(1)}, \ldots, r_i^{(K_s)}\}$ that are most semantically similar to $r_i$ and are present as outgoing edges from $e_{i-1}$ and then proceed with these alternatives. This approach allows us to construct approximate yet valid subgraphs grounded in the KG, thereby enhancing robustness when exact relation paths are unavailable.

## 4.4 ANSWER GENERATION

In the final stage, the LLM conducts faithful reasoning and generates the final answer based on the retrieved subgraph $\mathcal{G}_q$. Although $\mathcal{G}_q$ is compact and contextually relevant, it may still contain distracting facts. We therefore employ an LLM as a final reasoner to interpret this evidence. Specifically, we construct a prompt containing the original question $q$ and a linearized representation of the subgraph $\mathcal{G}_q$, which includes the successful reasoning paths connecting $e_0$ to potential answer entities. By conditioning on this structured evidence, the LLM is guided to generate an answer that is factually grounded in the KG. The complete inference process is summarized in Appendix A.2.

## 5 EXPERIMENTS

### 5.1 EXPERIMENTAL SETUP

**Datasets.** We evaluate RCR on two standard multi-hop KGQA benchmarks: WebQuestionsSP (WebQSP) (Yih et al., 2016) and Complex WebQuestions (CWQ) (Talmor & Berant, 2018). Both are built upon Freebase (Bollacker et al., 2008), with CWQ demanding more complex compositional reasoning (up to 4-hops) than WebQSP (up to 2-hops). Following prior works (Sui et al., 2024; Long et al., 2025; Luo et al., 2023; Li et al., 2024b), we adopt the preprocessed versions of the datasets, where question-specific subgraphs are pre-extracted based on entity linking.

**Evaluation Metrics.** Evaluation of RCR is two-fold, covering both the final KGQA performance and the intermediate relation and knowledge retrieval quality. For the main KGQA task, we follow standard evaluation protocols and report Hits@1 and F1. Hits@1 measures the proportion of questions whose top-1 answer is correct, while F1 computes the harmonic mean of precision and recall over the predicted answer sets. For evaluating the relation retrieval module, we employ Relation Retrieval Accuracy (RRA), which measures the percentage of questions for which all relations in the ground-truth reasoning paths are correctly retrieved. For the path generation module, we use Path Recall (PR), defined as the proportion of questions for which at least one generated relation path exactly matches a ground-truth reasoning path.

**Implementation Details.** We encode questions and relations into 1024-dimensional embeddings using *gte-large-en-v1.5* (Li et al., 2023b). The relation retriever, implemented as a two-layer MLP, is trained for 50 epochs using the Adam optimizer (Kingma, 2014). The training process utilizes 80 and 100 negative samples per question for WebQSP and CWQ, respectively. For the relation path generation stage, we instruction-tune LLaMA-3.1-8B-Instruct (Meta, 2024) for 5 epochs, conditioning it on the 200 candidate relations for each question. During inference, we retrieve top-200 candidate relations per question and apply beam search with beam sizes of 4 (WebQSP) and 6 (CWQ) to generate relation paths. During the subgraph retrieval process, we set the similarity-based relation substitution to use the top-1 most similar relation. Finally, we use the RoG model (Luo et al., 2023) for the generation of responses, as it has shown strong performance in the KGQA tasks. All experiments were performed on 8 NVIDIA L40 GPUs, each with 48GB of memory.

### 5.2 KGQA PERFORMANCE

In this section, we present a comprehensive evaluation of RCR against a range of state-of-the-art KGQA methods, including LLM-only and KG-augmented LLM approaches. As summarized in Table 1, RCR establishes new SOTA performance across most metrics. On the WebQSP dataset, RCR achieves a Hits@1 of 88.9%, surpassing the previous best method by 1.1 %. Compared to other strong KG-RAG baselines, RCR demonstrates a notable improvement of 3.2% over RoG (Luo et al., 2023) and 2.3% over SubgraphRAG (Li et al., 2024b) in Hits@1. The advantages of our relation-centric paradigm are even more pronounced on the more challenging CWQ dataset, which requires deeper multi-hop reasoning. RCR achieves a Hits@1 of 66.7% and an F1 score of 57.3%, outperforming the previous best method by 4.1% in Hits@1 and 1.3% in F1. These substantial improvements validate our central hypothesis: by identifying a high-quality set of candidate relations, RCR provides the LLM with a structured and semantically rich search space. This enables the model to construct more accurate and coherent reasoning paths. The high-quality subgraphs retrieved by RCR supply the final LLM reasoner with rich and relevant context, directly contributing to more faithful and precise answer generation.

### 5.3 ABLATION AND ANALYSIS

In this section, we evaluate the core components of RCR: the relation retriever, the path generator, and the subgraph retrieval process. The performance of these modules is critical, as they are responsible for constructing the high-quality, evidence-rich subgraph that underpins the final answer generation.

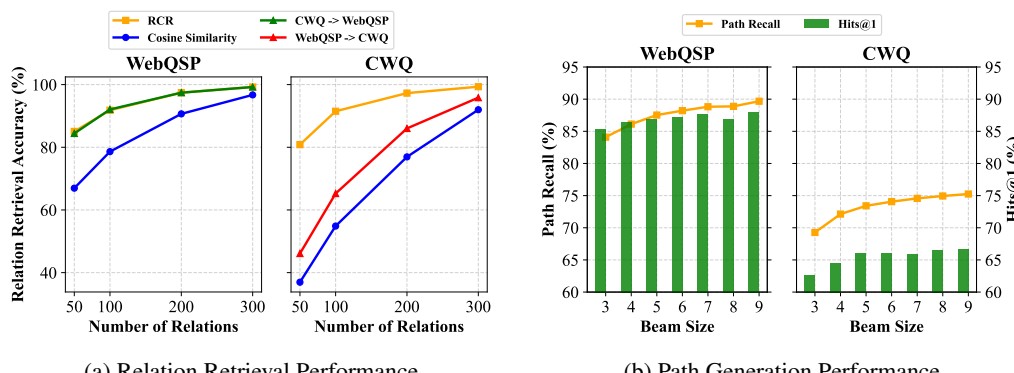

(a) Relation Retrieval Performance    (b) Path Generation Performance

Figure 2: Performance evaluation of RCR's core components on WebQSP and CWQ. (a) Relation Retrieval Accuracy under various numbers of retrieved relations ($k$). (b) Path Recall under different beam sizes used in path generation. The notation "A→B" indicates the relation retriever in RCR is trained on dataset A and tested on dataset B.

### 5.3.1 RELATION RETRIEVER

**Design of Relation Retriever.** We perform an ablation study to evaluate the design choices of our relation retriever, focusing on the hidden dimensions of the MLP and the number of negative samples used during training, as shown in Table 2. The results indicate that a hidden dimension of 1024 achieves the highest RRA when paired with 80 negative samples, suggesting that this configuration effectively balances model capacity and generalization. Increasing the hidden dimension to 1920 provides comparable performance but may introduce additional computational overhead.

Table 2: Ablation study on the design of the relation retriever. RRA (%) on the WebQSP validation dataset under 200 retrieved relations is reported.

| Hidden Dim. | Negative Samples | | |
|---|---|---|---|
| | 80 | 100 | 120 |
| 1024 | **93.58** | 92.30 | 92.30 |
| 1920 | 93.16 | 92.73 | **93.58** |

**Evaluation for Relation Retrieval.** The first step of our framework retrieves a set of candidate relations relevant to the input question. The quality of this set is fundamental to the success of our approach, as it provides the necessary foundation for the subsequent processes. As shown in Figure 2a, compared to a standard baseline that relies solely on pre-trained embeddings and cosine similarity, our lightweight MLP-based retriever demonstrates superior performance in identifying the ground-truth relations, especially at smaller retrieval sizes ($k = 50$). It demonstrates that the trainable projection head effectively captures the nuanced semantic alignment between questions and KG relations, providing a compact and high-fidelity candidate set for the downstream path generator.

Furthermore, the retriever demonstrates strong generalization capabilities, as shown in Figure 2a. For instance, when trained on WebQSP and evaluated on the unseen CWQ dataset (WebQSP→CWQ), our model achieves an 86.0% RRA at $k = 200$, surpassing the cosine similarity baseline by 9.1%. This indicates that our retriever learns a robust and transferable semantic mapping from natural language to relational structures, a crucial property for adapting RCR to diverse KGQA scenarios. Based on these results, we set $k = 200$ for all subsequent experiments to balance recall and efficiency, as a larger $k$ introduces more noise and slows down path generation. Detailed numerical results are provided in Appendix A.4.

### 5.3.2 PATH GENERATOR

Given the retrieved candidate relations, the next critical step is to generate coherent reasoning paths that connect the topic entity to the answer entities. This task is inherently challenging, as it requires

Table 3: Impact of the beam size on Hits@1 (%) and F1 (%) on WebQSP and CWQ, using a fixed candidate relation size of 200. $\leftrightarrow$ indicates path generator trained on one dataset and tested on another.

| Beam Size | WebQSP | | CWQ | |
|---|---|---|---|---|
| | Hits@1 (%) | F1 (%) | Hits@1 (%) | F1 (%) |
| 4 | 86.42 | **70.04** | 64.40 | 56.94 |
| 5 | 86.85 | 69.26 | 66.01 | 57.60 |
| 6 | **87.16** | 69.73 | **66.07** | **57.63** |
| 4 ($\leftrightarrow$) | 82.61 | 63.77 | 50.09 | 42.64 |
| 5 ($\leftrightarrow$) | 84.33 | 64.03 | 51.11 | 43.45 |
| 6 ($\leftrightarrow$) | 83.90 | 63.81 | 51.43 | 43.58 |

synthesizing multiple relations into a semantically meaningful sequence. As shown in Figure 2b, our LLM-based path generator shows a strong correlation between the beam size used for generation and the quality of the resulting paths, which directly impacts the final KGQA performance.

The impact of beam size on final performance is quantified in Table 3. On both datasets, a larger beam size correlates with improved Hits@1 scores. For instance, on the challenging CWQ dataset, increasing the beam size from 4 to 6 raises the Hits@1 score by 1.67 % (from 64.40% to 66.07%). This trend validates our approach, confirming that a wider search allows the LLM to construct better reasoning paths. The consistent improvement in KGQA metrics underscores the effectiveness of our path generation strategy in providing a strong foundation for the final answer generation. Consequently, we adopt beam sizes of 4 for WebQSP and 6 for CWQ in our main experiments to optimize the trade-off between performance and efficiency.

### 5.3.3 SIMILARITY-BASED RELATION SUBSTITUTION

To evaluate the impact of the similarity-based relation substitution mechanism, we conduct an ablation study by varying the number of top similar relations used for substitution ($K_s$). The results are summarized in Table 4. When no substitution is applied ($K_s = 0$), RCR achieves Hits@1 scores of 87.16% on WebQSP and 66.07% on CWQ. Introducing substitution with the top-1 most similar relation ($K_s = 1$) improves Hits@1 to 88.94% on WebQSP and 66.69% on CWQ, demonstrating the effectiveness of leveraging similar relations to address KG incompleteness. Further increasing $K_s$ to 2 yields the best Hits@1 performance on both datasets, with 89.31% on WebQSP and 66.83% on CWQ. However, this comes at the cost of a slight drop in F1 scores, indicating that while additional substitutions can enhance recall, they may introduce noise that affects precision. We therefore select $K_s = 1$ as our default setting, balancing the trade-off between Hits@1 and F1.

Table 4: Ablation study on the impact of the number of top similar relations used for substitution ($K_s$). Hits@1 and F1 scores are reported for WebQSP and CWQ, using a fixed candidate relation size of 200 and a beam size of 6. The best results are highlighted in **bold**.

| Substitutions (top-$K_s$) | WebQSP | | CWQ | |
|---|---|---|---|---|
| | Hits@1 (%) | F1 (%) | Hits@1 (%) | F1 (%) |
| 0 (w/o substitution) | 87.16 | **69.73** | 66.07 | **57.63** |
| 1 | 88.94 | 67.67 | 66.69 | 57.27 |
| 2 | **89.31** | 65.92 | **66.83** | 56.23 |

## 6 CONCLUSION

We introduced RCR, a new framework that abandons the noisy entity space and instead uses the stable, semantically rich space of relations as the foundation for reasoning. By identifying and composing relevant relations into reasoning paths, RCR extracts compact and coherent subgraphs, avoiding the ambiguity of traditional methods. Our approach establishes a new SOTA on multi-hop KGQA benchmarks, significantly improving reasoning accuracy and interpretability.

## ETHICS STATEMENT

This research is dedicated to advancing foundational scientific challenges in knowledge-based Artificial Intelligence. Our work is purely computational and adheres to high standards of research integrity. Our experiments were conducted on established, publicly available datasets (WebQSP, CWQ), and we did not use any sensitive, private, or proprietary data. The language models and encoders used in our framework are open-source and publicly available for research, and we have adhered to their usage policies.

## REPRODUCIBILITY STATEMENT

To ensure the reproducibility of our work, we have provided a detailed description of our method in Section 5.1. Appendix A.5 provides comprehensive implementation details, including dataset statistics, configurations, and hyperparameter settings. We will make our source code and model weights publicly available upon acceptance to facilitate further research and verification of our results.

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

# A APPENDIX

## A.1 THE USE OF LARGE LANGUAGE MODELS

We utilized a large language model (LLM) as a writing assistant to improve grammar, clarity, and polish the language. However, all scientific contributions, including the core ideas, experimental design, and interpretation of results, are the work of human authors only. The authors have critically reviewed and edited all content and take full responsibility for the final manuscript.

## A.2 FULL INFERENCE PIPELINE

The full inference pipeline for our proposed method, RCR, is detailed in Algorithm 1. Given a question $q$ and a topic entity $e_0$, the algorithm first retrieves a set of candidate relations using the `Relation Retriever`. Next, it generates potential relation paths with the `Relation-Centric Reasoning` module. The algorithm then traverses the knowledge graph (KG) according to these paths, applying path correction when necessary via the `Similarity-based Relation Substitution` mechanism. Finally, it synthesizes the final answer using the `Answer Generator` based on the constructed subgraph.

---

**Algorithm 1** RCR Inference Pipeline

---

**Input**: KG $\mathcal{G}$, question $q$, topic entity $e_0$
**Output**: Final answer $\mathcal{A}_q$
1: $\mathcal{R}_q \leftarrow$ RelationRetriever$(q, \mathcal{R}, K_r)$ {Retrieve top-K relevant relations}
2: $\mathcal{P}_q \leftarrow$ RelationReasoning$(q, e_0, \mathcal{R}_q)$ {Generate potential relation paths}
3: Initialize subgraph $\mathcal{G}_q \leftarrow \emptyset$
4: **for** each relation path $\mathcal{P}^j = [r_1, r_2, \ldots, r_T] \in \mathcal{P}_q$ **do**
5:      $e_0^j \leftarrow e_0$
6:      **for** each relation $r_t \in \mathcal{P}^j$ **do**
7:         **if** no triple $(e_{t-1}^j, r_t, *) \in \mathcal{G}$ **then**
8:            $\mathcal{R}_t' \leftarrow$ FindSimilarRelation$(e_{t-1}^j, r_t, \mathcal{R}_q, K_s)$ {Find similar relations if path is broken}
9:         **else**
10:            $\mathcal{R}_t' \leftarrow \{r_t\}$
11:         **end if**
12:         Initialize $\mathcal{E}_t^j \leftarrow \emptyset$
13:         **for** each $r \in \mathcal{R}_t'$ **do**
14:            $T \leftarrow \{(e_{t-1}^j, r, e_t) \in \mathcal{G}\}$
15:            $\mathcal{G}_q \leftarrow \mathcal{G}_q \cup T$
16:            Add all $e_t$ from triples in $T$ to $\mathcal{E}_t^j$
17:         **end for**
18:         $e_t^j \leftarrow$ Select$(\mathcal{E}_t^j)$
19:      **end for**
20: **end for**
21: $\mathcal{A}_q \leftarrow$ AnswerGenerator$(q, e_0, \mathcal{G}_q)$ {Generate final answer from the subgraph}
22: **return** $\mathcal{A}_q$

---

## A.3 COMPLETE PROMPTS FOR RELATION-CENTRIC REASONING

Figure 3 shows the prompt used for the `Relation-Centric Reasoning` module, where the placeholders `<Question>`, `<Entity>`, and the list of relations are filled in at runtime. This prompt is designed to guide the LLM in generating coherent and relevant relation paths based on the provided question and candidate relations.

## A.4 DETAILED RESULTS OF THE RELATION RETRIEVER

Table 5 provides a detailed quantitative analysis of the relation retriever's performance, complementing the trends shown in Figure 2a. Our RCR model consistently and substantially outperforms the

> **Prompt for Relation-Centric Reasoning**
>
> Please generate a valid relation path that connects the question entity to potential answer entities using only the provided candidate relations.
>
> Question: `<Question>`
> Question Entity: `<Entity>`
> Candidate Relations: `(<Relation 1>, <Relation 2>, ..., <Relation K>)`
>
> Requirements:
> - Only use relations from the candidate set.
> - Do not invent new relations.
> - Prefer shorter and semantically coherent paths.
> - Return the final path as a list of relations in order, e.g., (relation1, relation2, relation3).

Figure 3: The complete prompt template used for the `Relation-Centric Reasoning` module.

standard cosine similarity baseline across both datasets. The advantage is particularly pronounced when retrieving a smaller number of candidate relations ($k = 50$), where RCR achieves an RRA of 80.8% on CWQ, more than doubling the baseline's 36.9%. This highlights the effectiveness of our trained retriever in accurately identifying relevant relations even within a compact candidate set.

Furthermore, the table underscores the strong generalization capabilities of our retriever. When trained on the more complex CWQ dataset and tested on WebQSP, the model's performance is nearly identical to that of the model trained natively on WebQSP, indicating that it learns features applicable to simpler reasoning tasks. Conversely, while there is a performance drop when generalizing from the simpler WebQSP to the more complex CWQ, the model still surpasses the cosine similarity baseline by a significant margin (e.g., 86.0% vs. 76.9% at $k = 200$). These results confirm that our relation retriever learns a robust and transferable mapping from questions to relations.

Table 5: Relation Recall Accuracy (RRA) Performance Comparison. 'RCR (Generalization)' indicates the model is trained on one dataset and tested on another.

| Method | Dataset | Number of Relations | | | |
|---|---|---|---|---|---|
| | | 50 | 100 | 200 | 300 |
| Cosine Similarity | CWQ | 0.369 | 0.548 | 0.769 | 0.919 |
| | WebQSP | 0.669 | 0.786 | 0.906 | 0.967 |
| RCR | CWQ | 0.808 | 0.914 | 0.972 | 0.993 |
| | WebQSP | 0.850 | 0.918 | 0.974 | 0.992 |
| RCR (Generalization) | WebQSP → CWQ | 0.460 | 0.652 | 0.860 | 0.958 |
| | CWQ → WebQSP | 0.844 | 0.920 | 0.974 | 0.992 |

## A.5 MORE DETAILED EXPERIMENTAL SETUP

### A.5.1 DATASET.

We conduct experiments on two widely used complex question answering datasets: Complex WebQuestions (CWQ) (Talmor & Berant, 2018) and WebQuestionsSP (WebQSP) (Yih et al., 2016). We follow previous works (Jiang et al., 2022; Luo et al., 2023) to use the same train and test splits for fair comparison. Table 6 provides statistics on the number of questions. Table 7 shows the distribution of questions based on the number of hops required to answer them. CWQ contains a significant portion (20.75%) of questions that require three or more hops, highlighting its complexity compared

to WebQSP. Following Luo et al. (2023), we use the subgraph of Freebase constructed by extracting all triples that contain within the max reasoning hops of question entities in WebQSP and CWQ.

Table 6: Statistics of the question in the WebQSP and CWQ.

| Datasets | Train | Valid | Test | Max hop |
|----------|-------|-------|------|---------|
| WebQSP | 2,826 | 246 | 1,628 | 2 |
| CWQ | 27,639 | 3519 | 3,531 | 4 |

Table 7: Statistics of the question hops in WebQSP and CWQ.

| Dataset | 1 hop | 2 hop | $\geq$ 3 hop |
|---------|-------|-------|--------------|
| WebQSP | 65.49% | 34.51% | 0.00% |
| CWQ | 40.91% | 38.34% | 20.75% |

### A.5.2 TRAINING OF RELATION RETRIEVER.

We use the publicly available `gte-large-en-v1.5` (Li et al., 2023b) to generate initial 1024-dimensional embeddings for both questions and the textual descriptions of relations. On top of the frozen base encoder, we add trainable projection heads for both the question and relation encoders. Each head is a two-layer MLP with a SiLU activation function in between. The architecture is `Linear(1024 → 1024) → SiLU → Dropout(0.1) → Linear(1024 → 1024)`. The $\gamma$ in contrastive loss is set to 0.5. We train the relation retriever using the Adam optimizer with a learning rate of $5 \times 10^{-4}$ on WebQSP and $1 \times 10^{-3}$ on CWQ. Batch size is set to 1 since each question has a varying number of positive relations. The retriever is trained for 50 epochs on both datasets, and the model with the best RRA on the validation set is selected for inference.

### A.6 CASE STUDIES

---

**WebQTest-22**

**Question:**
what kind government does egypt have?

---

**Retrieved Relations:**
(location.country.form_of_government)
(government.form_of_government.countries)
(government.political_ideology.political_parties)
(government.political_party.ideology)
...

---

**Relation-Centric Reasoning (Beam 3):**
(government.form_of_government.countries)
(location.country.form_of_government)
(location.country.form_of_government, government.form_of_government.countries)

---

**Retrieved Knowledges:**
(Egypt → government.form_of_government.countries → Semi-presidential system)
(Egypt → government.form_of_government.countries → Provisional government)
...

---

**Answer Generation:**
Semi-presidential system
Provisional government

---

**Ground-truth:**
Semi-presidential system
Provisional government

---

## WebQTest-34

**Question:**
who did cam newton sign with?

**Retrieved Relations:**
(sports.pro_athlete.teams)
(sports.sports_team_roster.team)
(sports.sports_team_roster.player)
(book.book_subject.works)
(base.schemastaging.athlete_salary.team)
(american_football.football_player.position_s)
(base.schemastaging.athlete_salary.athlete)
...

**Relation-Centric Reasoning (Beam 4):**
(base.schemastaging.athlete_salary.athlete", "base.schemastaging.athlete_salary.team)
(base.schemastaging.athlete_extra.salary", "base.schemastaging.athlete_salary.team)
(sports.sports_team_roster.player, sports.sports_team_roster.team)
(sports.pro_athlete.teams, sports.sports_team_roster.team)

**Retrieved Knowledges:**
(Cam Newton → base.schemastaging.athlete_salary.athlete → m.010dqbnl → base.schemastaging.athlete_salary.team →
Carolina Panthers)
(Cam Newton → base.schemastaging.athlete_extra.salary → m.0j37n7l → base.schemastaging.athlete_salary.team
→ Carolina Panthers)
(Cam Newton → sports.sports_team_roster.player → m.0hpgnsj → sports.sports_team_roster.team → Florida Gators football)
(Cam Newton → sports.sports_team_roster.player → m.0z23kt0 → sports.sports_team_roster.team → Auburn Tigers football)
(Cam Newton → sports.pro_athlete.teams → m.04nb7yn → sports.sports_team_roster.team → Carolina Panthers)

**Answer Generation:**
Carolina Panthers

**Ground-truth:**
Carolina Panthers

## WebQTrn-124_f3990dc9aa470fa81ec4cf2912a9924f

**Question:**
Which movie with a character called Ajila was directed by Angelina Jolie?

**Retrieved Relations:**
(film.actor.film)
(film.performance.actor)
(film.performance.film)
(film.film_character.portrayed_in_films)
(film.film.starring)
(film.performance.character)
(fictional_universe.fictional_character.gender)
(film.film.country)
...

**Relation-Centric Reasoning (Beam 6):**
(film.film_character.portrayed_in_films", film.performance.film)
(film.performance.character", film.performance.film)
(film.editor.film)
(film.director.film)
(film.film_character.portrayed_in_films, film.film.starring)
(film.performance.character, film.film.starring)

**Retrieved Knowledges:**
(Angelina Jolie → people.person.gender → Female → fictional_universe.fictional_character.gender → Lara Croft)
(Angelina Jolie → film.editor.film → Unbroken", "Angelina Jolie → film.director.film → In the Land of Blood and Honey",
"Angelina Jolie → film.director.film → By the Sea)
(Angelina Jolie → film.director.film → A Place in Time)
(Ajla → film.performance.character → m.0gw7h9w → film.performance.film → In the Land of Blood and Honey)
(Ajla → common.topic.article → m.0gwfc1x)
(Ajla → film.performance.character → m.0gw7h9w → film.performance.actor → Zana Marjanovic)

**Answer Generation:**
In the Land of Blood and Honey

**Ground-truth:**
In the Land of Blood and Honey

## WebQTrn-62_16d8eb52603fa2d47a3441e9a5231568

**Question:**
Which child of Walt Disney died from lung cancer?

**Retrieved Relations:**
people.deceased_person.cause_of_death
people.person.children
people.person.parents
medicine.disease.notable_people_with_this_condition
people.cause_of_death.people
people.cause_of_death.parent_cause_of_death
...

**Relation-Centric Reasoning (Beam 6):**
(people.deceased_person.cause_of_death)
(people.person.children)
(people.person.parents)
(people.cause_of_death.people)
(people.person.nationality, people.person.nationality)
(fictional_universe.fictional_character.gender, people.person.gender)

**Retrieved Knowledges:**
(Lung cancer → people.deceased_person.cause_of_death → Sharon Mae Disney)
(Lung cancer → people.deceased_person.cause_of_death → Walt Disney)
(Walt Disney → people.deceased_person.cause_of_death → Lung cancer)
(Walt Disney → people.person.children → Diane Disney Miller)
(Walt Disney → people.person.parents → Flora Call Disney)
(Walt Disney → people.person.parents → Elias Disney)
(Walt Disney → people.person.parents → Sharon Mae Disney)
(Walt Disney → people.cause_of_death.people → Circulatory collapse)

**Answer Generation:**
Sharon Mae Disney

**Ground-truth:**
Sharon Mae Disney

