# OpenReview forum: "RCR: Relation-Centric Reasoning with Large Language Models for Knowledge-based Question Answering"
_ICLR.cc/2026/Conference — Submitted to ICLR 2026_

### Official Review · Reviewer_3tSH · 2025-10-30

**Soundness:** 2
**Presentation:** 3
**Contribution:** 1
**Rating:** 2
**Confidence:** 5

**Summary:**

This paper proposes Relation-Centric Reasoning (RCR), a new paradigm for KGQA that shifts the focus from noisy entity-level expansion to relation-level reasoning. The framework first retrieves a small set of candidate relations relevant to the question, then prompts a fine-tuned LLM to generate relation paths constrained to this set. These abstract paths are instantiated into evidence subgraphs via a path-guided retrieval process, with a similarity-based relation replacement mechanism to handle KG incompleteness. Finally, the LLM generates the answer conditioned on the question and the evidence subgraph. Experiments on WebQSP and CWQ show significant improvements over prior state-of-the-art in both accuracy and robustness, while also offering more interpretable reasoning traces.

**Strengths:**

1. The paper gives a novel perspective for KGQA. Shifting from entity-centric to relation-centric reasoning addresses noise and inefficiency in KG-RAG, which is conceptually appealing.
2. The paper gives a clear pipeline design: The four-stage framework (relation retrieval → relation-path generation → path-guided subgraph retrieval → answer generation) is logically structured and interpretable.
3. RCR achieves new state-of-the-art results on both WebQSP and CWQ, with particularly notable gains on multi-hop reasoning.

**Weaknesses:**

1. The approach heavily relies on relation retrieval. If the gold relation is missed in top-K, the pipeline collapses.
2. It assumes that the generated relation path must logically match the KG path, but in practice, many valid answers can come from semantically different paths, reducing reliability.
3. Experiments are limited to WebQSP and CWQ. Scalability to larger, noisier KGs in other domains remains unclear.
4. Relation retrieval might still be costly in very large KGs with hundreds of thousands of relations. Both path generation and answer generation rely on fine-tuned LLMs, which adds cost and reduces reproducibility.

**Questions:**

1. How robust is RCR to cases where valid answer paths do not exactly match the “logical relation sequence” in the KG? Can semantically equivalent but structurally different paths still lead to correct answers?
2. How does relation replacement affect precision? Does it sometimes introduce spurious reasoning by replacing with semantically “similar” but logically irrelevant relations?

---

### Official Review · Reviewer_Ymms · 2025-10-31

**Soundness:** 3
**Presentation:** 3
**Contribution:** 3
**Rating:** 4
**Confidence:** 3

**Summary:**

The paper proposes RCR (Relation-Centric Reasoning) for knowledge graph question answering, which uses a dual-encoder relation retriever and LLM to get and compose relations into relation paths (or abstract reasoning plans). The paths are then grounded on the KG to retrieve a subgraph, based on which a final LLM generates the answer. Experiments show state-of-the-art performance and the effect of technical components.

The paper is clearly written and demonstrates solid empirical improvements on WebQSP and CWQ. The relation-centric framing is intuitively appealing and produces interpretable reasoning traces. However, the novelty appears incremental, and the core claims—that relation-centric reasoning is fundamentally advantageous and more scalable—are supported indirectly rather than through controlled comparisons or empirical cost analysis. While the results are promising, the contribution may fall short of the bar for ICLR in terms of conceptual originality and depth of validation. With stronger evidence isolating the benefit of the relation-first design, or a more rigorous analysis of scalability, the paper would be more compelling.

**Strengths:**

1. The paper introduces a relation-centric framework that departs from conventional triple-level scoring and entity-based retrieval, which are prone to noise and ambiguity in large KGs.
2. Relation paths and subgraphs can be inspected; case studies illustrate interpretable and coherent multi-hop reasoning paths grounded in the KG.
3. The similarity-based substitution mechanism improves robustness when predicted relations are absent from the KG, helping maintain valid reasoning paths.

**Weaknesses:**

1. Novelty may be incremental. Although the relation-centric perspective is interesting, many prior works also constrain LLM reasoning with relational constraints (RoG, StructGPT, ReasoningLM, Fidelis). The core components—dual-encoder retrieval, LLM-based path composition, and KG grounding—are adapted from existing techniques. The contribution lies more in integrating these pieces into a cleaner pipeline than in introducing a fundamentally new reasoning mechanism.
2. Lack of a direct entity-centric vs relation-centric control experiment. Ablations on its components support the effectiveness of the relation-centric design. Still, there is no controlled comparison to an analogous entity-centric variant under the same architecture to isolate the specific benefit of relation-level reasoning.
3. Scalability claims could be better substantiated. The design—fixed top-K relation vocabulary and only two LLM calls—suggests improved computational efficiency. However, the paper does not report end-to-end inference time or cost trends with respect to the size of the relation vocabulary. Even a small latency study or scaling trend would make the scalability argument more explicit.

**Questions:**

na

---

### Official Review · Reviewer_o2Xc · 2025-11-01

**Soundness:** 3
**Presentation:** 3
**Contribution:** 3
**Rating:** 4
**Confidence:** 4

**Summary:**

This paper proposes Relation-Centric Reasoning (RCR), a framework for knowledge-based question answering that shifts reasoning from the noisy entity space of knowledge graphs to the relation space. RCR first retrieves a small set of relevant relations for a question, then uses a large language model to compose these relations into multi-hop reasoning paths, grounds the paths in the KG through a similarity-based substitution mechanism, and finally generates an answer based on the retrieved subgraph. Experiments on WebQSP and ComplexWebQuestions show that RCR achieves state-of-the-art performance and improves interpretability and robustness compared to existing KG-augmented LLM methods.

**Strengths:**

1. Moving from entity/triple-centric retrieval to relation-centric reasoning is an elegant idea, helping address noise issues in KGQA.
2. The paper introduces a well-structured framework. The four-stage pipeline (retrieval, reasoning, subgraph grounding, and answering) is modular and clear.
3. I appreciate that the author list out clear training objectives (contrastive retrieval loss, NLL for path generation) in the main text.

**Weaknesses:**

1. This paper only cover limited dataset scope and diversity, rather thane the extensive evaluations as it claims. It only evaluates on two Freebase-based datasets (WebQSP and CWQ) and remains unclear how well the approach generalizes to other large KGs.
2. The paper has limited efficiency analysis to support its claim. The claim of “efficiency and scalability” is not quantitatively supported (e.g., latency, retrieval cost, number of LLM tokens).
3. The experiment results does not provide standard deviation or multi-run results. It will be challenging to identify if the 1% improvement in WebQSP Hit and CWQ F1 is statistically significant or just from random fluctuations.

**Questions:**

Please address the weakness mentioned above.

---

### Meta-Review · Area_Chair_RTh5 · 2025-12-09

**Summary:**

All three reviewers acknowledge the conceptual appeal of pivoting from entity- to relation-centric retrieval, yet converge on three fatal weaknesses:

1.  The pipeline integrates existing components (dual-encoder retrieval, LLM path generation, KG grounding) without a fundamentally new mechanism; prior works (RoG, StructGPT, ReasoningLM) already constrain LLMs with relational knowledge (R2, R3).
2. No ablation that isolates the relation-first design from an equivalent entity-centric baseline under the same architecture, leaving the claimed superiority unsupported (R2, R3).
3. Experiments are confined to two Freebase datasets (WebQSP, CWQ); no latency/memory scaling curves, no transfer to larger/noisier KGs, no variance reported, and gains of ≈1 % are indistinguishable from random fluctuation (R1–R3).

Further, R3 warns that top-K relation retrieval acts as a single point of failure. If gold relations are missed, the entire pipeline collaps—a fragility not mitigated in the paper.

**Reviewer Concerns:**

All concerns above are not addrssed, the authors did not provide a rebuttal.

**Reviewer Scores:**

R1 (4 → 3): No rebuttal provided.
R2 (4 → 3): No rebuttal provided.
R3 (2 → 2): no movement; structural fragility remains.

---

### Decision · Program_Chairs · 2026-01-26

Reject